# Effects of Physical Activity Level on Attentional Networks in Young Adults

**DOI:** 10.3390/ijerph19095374

**Published:** 2022-04-28

**Authors:** Fanying Meng, Chun Xie, Fanghui Qiu, Jiaxian Geng, Fengrong Li

**Affiliations:** 1Institute of Physical Education, Huzhou University, Huzhou 313000, China; 02730@zjhu.edu.cn; 2Department of Physical Education, Shanghai Jiao Tong University, Shanghai 200240, China; xiechun628@sjtu.edu.cn; 3Department of Physical Education, Qingdao University, Qingdao 266071, China; qiufanghui@qdu.edu.cn; 4Department of Physical Education and Military, Anhui University, Hefei 230601, China

**Keywords:** physical activity, attentional networks, young adults, sedentary behavior

## Abstract

Although physical activity is associated with better attentional functioning in elderly populations or in specific clinical populations, the association between physical activity level and attention has been less studied in young adult populations. Thus, the purpose of this study was to investigate whether the positive effects of physical activity on attentional networks extend to young adults. In total, 57 college students were recruited and assigned to one of three groups of physical activity levels (high, moderate, and low) based on their self-reported exercise. Each participant completed the Attention Network Test to evaluate the efficiency of three components of attention: alerting, orienting, and executive control. Compared with the low physical activity group, both the high and moderate physical activity groups exhibited better executive control. In addition, the efficiency of the executive control network was positively correlated with physical activity. By contrast, no statistically significant differences were detected among these three groups for the functioning of the alerting or orienting networks. These findings suggested that physical activity had a positive effect on attention in young adults, with the benefit primarily observed for the executive control component rather than for the alerting and orienting components of attention.

## 1. Introduction

Attention is the heart of the cognitive system, enabling humans to selectively concentrate on the most relevant information while ignoring irrelevant information [1,2]. It is also an indispensable element for a successful life because of its role in maintaining and organizing other cognitive functions (e.g., perception, memory, and thinking) [3,4]. In light of the importance of attention, researchers have increasingly focused on how to improve attention. Many studies have found that people with high levels of physical activity (PA) perform better on measures of attention than people with low levels [1,5,6,7,8]. Recent meta-analyses results indicate that PA intervention may be an effective approach for enhancing attention [9,10]. However, the existing studies have mainly assessed attention in children, the elderly, or specific clinical populations rather than in young adults, such as college students. Maintaining good attention is critical for enhancing academic performance in college students [9]. However, this population was shown to demonstrate impaired attention owing in part to smartphone addiction [11,12], substance abuse [13,14], and insufficient sleep [15,16].

Early investigators viewed human attention as a single system, but more recent work has shown that the theoretical models of a single system could not fully explain attention processing. Posner and Petersen [17] proposed the attention network theory, suggesting that human attention encompasses three functionally and anatomically independent networks, namely, an alerting network, orienting network, and executive control network. The alerting network, which is associated with the frontal and parietal cortices as well as with thalamic nuclei, ensures that individuals maintain a vigilant and alert state to incoming stimuli to enable rapid perception and processing at later stages. The orienting network, associated with the superior parietal lobe, temporal parietal junction, and frontal eye fields, enables individuals to select useful information from a wide range of sensory inputs. The executive control network, which is associated with the frontal cortex and anterior cingulate gyrus, is responsible for behavioral control and problem-solving under conflict conditions [17,18,19,20]. To accurately and effectively measure the attentional networks, Fan developed the Attention Network Test (ANT) based on a spatial cueing task and a flanker task [21]. The ANT measures efficiency in the alerting, orienting, and executive control systems independently through reaction time (RT) differences under various conditions and is widely used for assessments of healthy people as well as of people with attention disorders [22,23,24,25,26,27,28,29].

Assessments of PA typically include a single bout or repeated bouts of activity. A single bout of PA was shown to influence the performance of attention tasks shortly after the PA intervention. Regarding the effects of a single bout of PA on the attention network, researchers have suggested that the efficiency of the alerting network is increased during aerobic exercise [30], whereas after acute cycling exercise, the efficiency of the executive network is increased [31]. Although acute bouts of aerobic exercise exert a positive effect on the attention networks, the effects of chronic exercise must also be taken into account because of their long-lasting or even permanent changes in brain structure and function. Recent research has provided promising evidence that repeated bouts of PA can improve the three components of attentional networks in school children [32] and older adults [33,34]. It was also suggested that repeated bouts of PA enhance the connectivity of the attentional networks in older adults [35] and in patients with stroke [29]. However, the current knowledge regarding the effects of repeated bouts of PA on the three components of the attentional networks in young adults stems from only three reports. Using the ANT, Pérez et al. [36] investigated whether chronic exercise improves attention networks and found that compared with passive participants, active participants exhibited superior executive control. On the basis of these findings, Wang et al. [4] used the same method to explore the association between repeated bouts of PA and attention networks in athletes. They observed that table tennis athletes displayed enhanced performance in the executive control network. These recent studies provide preliminary evidence suggesting that there is a positive effect of PA selectively on the executive control network. However, this interpretation has recently been challenged in a study conducted by Johnstone and Marí-Beffa [25] demonstrating that the improvement of repeated bouts of PA on attention networks was observed only in the alerting network. Given these inconsistent results, the effects of repeated bouts of PA on the three components of the attention network require further exploration.

Relatively few studies have investigated the influence of PA on attentional networks in young adults, and the relevant findings are inconsistent [4,25,36]. Therefore, additional research assessing the influence of PA on attentional networks in young adults is required. In addition, the effects of PA levels on attentional networks are unknown; thus, it remains important to investigate the effects of PA levels on attentional networks in young adults. Therefore, the aim of the present study was to investigate whether the positive effects of PA on attention observed in other populations extend to young adults and whether PA level affects the attentional networks in young adults. College students were recruited to complete a demographic characteristics and PA survey and to perform the ANT. We hypothesized that PA has beneficiary effects on the three attention network components, especially on the executive control network. Thus, college students with higher PA levels would perform better on the ANT than students with lower PA levels.

## 2. Materials and Methods

### 2.1. Participants

According to the results of meta-analyses conducted by Haverkamp et al. [9] and by Xue et al. [37], the effect size for the ability of PA to improve cognitive function was approximately 0.2. Using this effect size, an alpha of 0.05, and 80% power, we calculated the sample size for the present study by using Gpower3.1 software. The results showed that a minimum of 54 participants were required. Thus, we recruited 57 college students from Huzhou University through advertisements posted at the university. Each included participant completed questionnaires describing their demographic characteristics and PA levels.

Students were assigned to a high PA group, a moderate PA group, or a low PA group based on their answers to the International Physical Activity Questionnaire (IPAQ, the short version in Chinese), which is widely used to evaluate PA and has high validity and reliability [38,39]. The participants were grouped by their self-reported MET-min/week in accordance with the guideline of the IPAQ as follows. Participants assigned to the high PA group were required to meet at least one of the following two criteria: (1) vigorous-intensity activity ≥ 3 days/week and an overall weekly PA ≥ 1500 MET-min/week; (2) any combination of light-intensity, moderate-intensity, and vigorous-intensity activities up to 7 days/week, with overall weekly PA ≥ 3000 MET-min/week. Participants assigned to the moderate PA group were required to meet any of the following three criteria: (1) at least 20 min of vigorous-intensity activity each day for ≥3 days/week; (2) at least 30 min of moderate-intensity activity or light-intensity activity for ≥5 days/week; (3) any combination of light-intensity, moderate-intensity, and vigorous-intensity activities ≥ 5 days/week, with overall weekly PA ≥ 600 MET-min/week. Participants who met either of the following two criteria were assigned to the low PA group: (1) reported no PA; (2) reported some PA but did not meet the criteria for the high or moderate PA group. All participants were right-handed, had normal or corrected-to-normal vision, and had no psychiatric disorder, brain injury, or neurologic illness. Each participant provided written informed consent before being tested and received a small gift or financial compensation for their time after the test. This study adhered to the ethical guidelines of the Declaration of Helsinki and was approved by the Ethics Committee of Huzhou University (Approval No. SGY-2021-07) [40].

### 2.2. ANT Task Procedure

On the basis of a previous study by Chang et al. [31], we programmed the ANT using the E-prime 2.0 software package (Psychology Software Tools, Pittsburgh, PA, USA). All Stimuli were presented on a 17-inch Lenovo computer monitor with a gray background that was placed in front of a participant at a distance of 60 cm. The sequence of the ANT and the timing of the events within a trial are shown in Figure 1.

Each trial started with the presentation of a fixation cross from 400 to 1600 ms in random order. Then, one of four possible cues appeared for 100 ms: no cue, a center cue, a double cue, or a spatial cue. For the no cue condition, only a fixation cross was presented. In the center cue condition, the fixation cross was replaced by an asterisk. In the double cue condition, two asterisks were presented, one above and one below the fixation cross. In the spatial cue condition, one asterisk was presented, either above or below the fixation cross. When the cue disappeared, the fixation cross continued to be presented for 400 ms. Thereafter, a response stimulus was presented above or below the fixation cross. The response stimulus consisted of a central target and four flankers, two flankers on either side of the central target. The relationship between the central target and the four flankers was divided into three conditions: congruent, incongruent, and neutral. In the congruent condition, both the central target and the four flankers were arrows that all pointed leftward or rightward, with the direction of the central target the same as that for the four flankers (i.e., →→→→→). In the incongruent condition, both the central target and the four flankers were also arrows that pointed either leftward or rightward, but the direction of the central target was different from that of the four flankers, which were all in the same direction (i.e., →→←→→). In the neutral condition, the central target was an arrow that pointed leftward or rightward, but the four flankers were lines without arrowheads (i.e., −−→−−). Participants were instructed to decide the direction of the central target as quickly and accurately as possible by pressing the corresponding key on a numeric keyboard. A central target with a left direction was assigned to the “1” key to be pressed with the left index finger, and a central target with a right direction was assigned to the “3” key to be pressed with the right index finger. Participants had to respond with a key press within 1700 ms, otherwise, the next trial was triggered.

To become familiar with the ANT task, each participant completed a practice experiment before the formal experiment began. The practice experiment consisted of 24 trials, which included all 12 possible cue-flanker combinations in a randomized order. Feedback on the response times and accuracy rates were given to the participants during this period. The formal experiment followed and consisted of 288 trials that were divided into six blocks. Within each block, the frequency of the cue and flanker conditions and their combinations were equal. No feedback was given during the formal experiment. It took approximately 15 min to complete the entire task.

### 2.3. Statistical Analysis

#### 2.3.1. Demographic Characteristics

To rule out the potential effects of demographic variables, a one-way analysis of variance (ANOVA) was performed to compare the demographic characteristics (gender, age, years of education, height, body mass, body mass index, PA level, and the amount of time spent sitting) among the high, moderate, and low PA groups.

#### 2.3.2. Reaction Times and Accuracy Rates

In order to reduce errors caused by extreme and incorrect values, RTs below 200 ms and above 1700 ms and incorrect trials were excluded from further analysis. Mean RT and accuracy rate were assessed using three-way repeated ANOVAs with within-participants factors of cue (none, central, double, and spatial) and flanker (congruent, incongruent, and neutral) and a between-participants factor of group (high, moderate, and low PA)

#### 2.3.3. Attention Networks

The three attention network effects were operationalized as follows: (1) alerting network effects were calculated as the RT difference between “no cue” and “double cue”; (2) orienting network effects were calculated as the RT difference between the “center cue” and “spatial cue”; and (3) executive control network effects were calculated as the RT difference between the “incongruent” and “congruent” conditions [21].

To investigate the effect of PA on the three attention networks, mean RT and accuracy rate for the alerting network were examined respectively in two-way repeated-measures ANOVAs, using a within-participants factor of cue (no cue and double cue) and a between-participants factor of group (high, moderate, and low PA). For the orienting network, two-way repeated-measures ANOVAs were performed to examine the mean RT and accuracy rate, using a within-participants factor of cue (center cue and spatial cue) and a between-participants factor of group (high, moderate, and low PA). For the executive control network, two-way repeated-measures ANOVAs were performed to examine the mean RT and accuracy rate, using a within-participants factor of flanker (congruent and incongruent) and a between-participants factor of group (high, moderate, and low PA).

One-way ANOVAs were used to further evaluate the effect of PA on the three attention network components. To achieve this aim, the following calculations were carried out: (1) alerting was the RT with no cue minus the RT with a double cue; (2) orienting was the RT with the center cue minus the RT with a spatial cue; (3) and executive control was the RT with an incongruent flanker minus the RT with a congruent flanker.

#### 2.3.4. Correlation Analysis

To further understand the relationship between the efficiency of the executive control network and the PA, a Pearson correlation analysis was performed.

## 3. Results

### 3.1. Demographic Characteristics

Table 1 presents the demographic characteristics of the included participants. No significant differences were observed among the three PA groups in terms of age (F_(2, 54)_ = 0.51, *p* = 0.61), years of education (F_(2, 54)_ = 0.85, *p* = 0.43), height (F_(2, 54)_ = 1.05, *p* = 0.36), body mass (F_(2, 54)_ = 0.31, *p* = 0.73), and body mass index (F_(2, 54)_ = 0.03, *p* = 0.97). As expected, a significant difference was obtained in PA level among the three groups (F_(2, 54)_ = 48.47, *p* < 0.001). A Dummetts T3 post-hoc comparison showed that the high PA group had a higher PA level than both the moderate and low PA groups (*ps* < 0.001), a significant difference was found in the PA levels between the moderate and low PA groups (*p* < 0.001). The IPAQ scores also revealed a significant difference in the amount of time spent sitting among the three groups (F_(2, 54)_ = 6.73, *p* < 0.001). An LSD post-hoc comparison showed that the moderate and low PA groups had longer sitting times than the high PA group (*ps* < 0.05), but the difference in sitting times between the moderate and high PA groups was not significant.

### 3.2. Reaction Times

The results of the 4 × 3 × 3 ANOVA assessing RT revealed statistically significant main effects of cue (F_(3, 52)_ = 77.08; *p* < 0.001; $\eta_{p}^{2}$ = 0.59) and flanker (F_(2, 53)_ = 804.96; *p* < 0.001; $\eta_{p}^{2}$ = 0.94).

The interaction between cue and flanker was also statistically significant (F_(6, 49)_ = 4.74; *p* < 0.001; $\eta_{p}^{2}$ = 0.08). A simple effects analysis of the interaction showed that for all cue conditions, participants responded faster on congruent conditions than incongruent conditions (*ps* < 0.001) and responded faster on neutral conditions than incongruent conditions (*ps* < 0.001). No significant differences were observed between congruent and neutral conditions (*ps* > 0.05) under no cue, central cue, and spatial cue conditions, but participants responded faster on the neutral condition than on the congruent condition under the double cue condition (*p* < 0.05). In addition, participants exhibited the shortest RT for the spatial cue and the longest RTs for the no cue under all flanker conditions.

The interaction between group and flanker was also statistically significant (F_(4, 108)_ = 2.95; *p* < 0.05; $\eta_{p}^{2}$ = 0.10). A simple effects analysis of the interaction showed that the high PA group responded faster than the low PA group on the incongruent condition (*p* < 0.05) and on the neutral condition (*p* < 0.05), but no significant differences were observed between these three groups for congruent condition. In addition, participants in all groups responded faster to congruent and neutral conditions than to incongruent conditions (*ps* < 0.001). No significant differences were observed between the congruent and neutral conditions for both the moderate and low PA groups, whereas participants in the high PA group responded faster to neutral conditions than to congruent conditions (*p* < 0.05).

The main effect of group did not reach statistical significance (F_(2, 54)_ = 2.06; *p* = 0.14; $\eta_{p}^{2}$ = 0.07). The interaction between group and cue also did not reach statistical significance (F_(6, 106)_ = 0.42; *p* = 0.81; $\eta_{p}^{2}$ = 0.02) nor did the interaction between group, cue, and flanker (F_(12, 100)_ = 0.73; *p* = 0.70; $\eta_{p}^{2}$ = 0.03) (Table 2).

### 3.3. Accuracy

The results of the 4 × 3 × 3 ANOVA assessing accuracy revealed statistically significant main effects of cue (*F*_(3, 52)_ = 13.89; *p* < 0.001; $\eta_{p}^{2}$ = 0.21) and flanker (F_(2, 53)_ = 42.43; *p* < 0.001; $\eta_{p}^{2}$ = 0.44).

The interaction between cue and flanker was also statistically significant (F_(6, 49)_ = 10.94; *p* < 0.001; $\eta_{p}^{2}$ = 0.17). A simple effects analysis of the interaction indicated that for all cue conditions, participants showed higher accuracy on congruent and neutral conditions than on incongruent conditions (*ps* < 0.001), whereas no statistically significant differences were observed between congruent and neutral conditions for any cue condition (*ps* > 0.05). In addition, participants exhibited the highest accuracy for the spatial cue and the lowest accuracy for the center cue under the incongruent condition (*ps* < 0.05). No significant differences were observed between each cue condition for congruent and neutral conditions (*ps* > 0.05).

The main effect of the groups did not reach statistical significance (F_(2, 54)_ = 2.14; *p* = 0.13; $\eta_{p}^{2}$ = 0.07). The interaction between group and flanker also did not reach statistical significance (F_(4, 108)_ = 1.67; *p* = 0.19; $\eta_{p}^{2}$ = 0.06) nor did the interaction between group and cue (F_(6, 106)_ = 0.54; *p* = 0.76; $\eta_{p}^{2}$ = 0.02) or the interaction between group, cue, and flanker (F_(12, 100)_ = 1.00; *p* = 0.43; $\eta_{p}^{2}$ = 0.04) (Table 3).

### 3.4. Alerting Network

The results of a 2 × 3 ANOVA revealed a statistically significant main effect of cue (F_(1, 54)_ = 67.28; *p* < 0.001; $\eta_{p}^{2}$ = 0.56). The main effect of the groups (F_(2, 54)_ = 1.87; *p* = 0.16; $\eta_{p}^{2}$ = 0.07) and the interaction between cue and group did not reach statistical significance (F_(2, 54)_ = 0.31; *p* = 0.73; $\eta_{p}^{2}$ = 0.01). Accuracy rates for the alerting network revealed a significant main effect of cue (F_(1, 54)_ = 18.94; *p* = 0.00; $\eta_{p}^{2}$ = 0.26). The main effect of the groups (F_(2, 54)_ = 1.20; *p* = 0.31; $\eta_{p}^{2}$ = 0.04) and the interaction between cue and group did not reach statistical significance (F_(2, 54)_ = 0.65; *p* = 0.53; $\eta_{p}^{2}$ = 0.02) (Figure 2a).

The results of a one-way ANOVA indicated that no differences were observed in the efficiency of the alerting network between the high, moderate, and low PA groups (F_(2, 54)_ = 0.31; *p* = 0.73) (Figure 2d).

### 3.5. Orienting Network

The results of a 2 × 3 ANOVA revealed a statistically significant main effect of cue (F_(1, 54)_ = 50.44; *p* < 0.001; $\eta_{p}^{2}$ = 0.48). However, the main effect of the group (F_(2, 54)_ = 1.92; *p* = 0.16; $\eta_{p}^{2}$ = 0.07) and the interaction between cue and group (F_(2, 54)_ = 0.68; *p* = 0.51; $\eta_{p}^{2}$ = 0.03) did not reach a statistical significance. The accuracy rates for the orienting network revealed a significant main effect of cue (F_(1, 54)_ = 11.35; *p* < 0.001; $\eta_{p}^{2}$ = 0.17) and of group (F_(2, 54)_ = 3.33; *p* < 0.05; $\eta_{p}^{2}$ = 0.11). However, the interaction between cue and group were not statistically significant (F_(2, 54)_ = 0.55; *p* = 0.56; $\eta_{p}^{2}$ = 0.02) (Figure 2b).

The results of a one-way ANOVA indicated no statistically significant differences in the efficiency of the orienting network among the high, moderate, and low PA groups (F_(2, 54)_ = 0.68; *p* = 0.51) (Figure 2d).

### 3.6. Executive Control Network

The results of a 2 × 3 ANOVA revealed a statistically significant main effect of flanker (F_(1, 54)_ = 1209.25; *p* < 0.001; $\eta_{p}^{2}$ = 0.96). The interaction between flanker and group was also statistically significant (F_(2, 54)_ = 6.19; *p* < 0.001; $\eta_{p}^{2}$ = 0.19). A simple effects analysis of the interaction showed that all groups responded faster to congruent conditions than to incongruent conditions (*ps* < 0.001). For the incongruent condition, the high PA group responded faster than the low PA group did (*p* < 0.05), but no significant differences were observed between the high and moderate PA groups or between the moderate and low PA groups (*p* > 0.05). The main effect of the group was not statistically significant (F_(2, 54)_ = 1.87; *p* = 0.16; $\eta_{p}^{2}$ = 0.07). Accuracy rates for the executive network revealed a statistically significant main effect of flanker (F_(1, 54)_ = 43.32; *p* < 0.001; $\eta_{p}^{2}$ = 0.45). However, the main effect of the group (F_(2, 54)_ = 1.89; *p* = 0.16; $\eta_{p}^{2}$ = 0.07) and the interaction between cue and group (F_(2, 54)_ = 1.78; *p* = 0.18; $\eta_{p}^{2}$ = 0.06) did not reach statistical significance (Figure 2c).

The results of a one-way ANOVA indicated that participants in the high PA group exhibited the smallest conflict effect, followed by participants in the moderate PA group, with participants in the low PA group exhibiting the largest conflict effect (F_(2, 54)_ = 6.19; *p* < 0.001) (Figure 2d).

### 3.7. Correlation between Executive Control and PA Level

The distribution of the data for executive control was normal (Kolmogorov–Smirnov = 0.11, *p* = 0.09). However, the data for the level of PA were not normally distributed (Kolmogorov–Smirnov = 0.20, *p* = 0.00). Thus, a logarithmic conversion was performed on the data for the PA level to fulfill the prerequisite for the use of Pearson correlation analyses. The results of the correlation analysis showed a significant correlation between executive control and the level of PA (r_(57)_ = −0.35, *p* < 0.01) (Figure 3), with a correlation coefficient higher than 0.3 considered a moderate correlation [41]. This finding indicated that the more PA an individual participated in, the less conflict effect they experienced and the higher the efficiency of their executive control network.

## 4. Discussion

The present study aimed to investigate the effects of the level of PA on attention networks (alerting, orienting, and executive control) in young adults. To this end, college students with different PA levels were recruited to participate in the ANT. Building on the results of the few studies that were conducted to determine the effects of PA on attentional networks, this is the first study to examine the influence of PA levels on attentional networks via ANT in young adults. Our primary results showed that, compared with the low PA group, both high and moderate PA groups exhibited higher efficiency of the executive control network, and the efficiency of this network was positively correlated with the level of PA. By contrast, there were no statistically significant differences among these three PA groups in the efficiency of the alerting and orienting networks. Overall, these findings provided additional evidence supporting previous research demonstrating that PA has a selective enhancement on attention networks [4,36].

### 4.1. PA and the Executive Control Network

Our study found that the efficiency of executive control in high and moderate PA groups was higher than that in the low PA group, whereas no significant difference was found between the high PA and moderate PA groups. This finding is consistent with that of Pérez et al. [36], who reported that physically active participants showed higher efficiency of their executive control network than physically passive participants. Our results, together with previous studies, further support the idea that PA has a positive effect on the executive control network. On the basis of whether a person’s daily activities promote health, overall PA could be separated into health-enhancing PA or sedentary behavior [42]. In the present study, the high PA group comprised young adults with a high level of PA combined with a short amount of time spent sitting, whereas the low PA group comprised young adults with a low level of PA and a long sitting time. Thus, higher health-enhancing PA and lower sedentary behavior might explain the superiority of the efficiency of the executive control network that was detected in the high PA group in the present study [43,44].

Numerous previous studies have suggested that PA has a beneficial effect on cognitive function. For instance, the results of a meta-analysis that included almost 100 studies suggested that PA has a positive predictive effect on cognitive function (e.g., executive control, memory, attention, and problem solving) at the behavioral level [9]. In addition, many imaging studies have also shown that long-term regular PA not only enlarges the volume of the white and grey matter [45,46,47] but also alters brain activity and functional connectivity of the anterior cingulate cortex, inferior frontal gyrus, and dorsolateral prefrontal cortex, all of which are highly involved with the executive control network [48,49]. Those changes in brain structure and function were considered to be the critical mechanisms underlying the improvement of regular PA on the executive control network. In addition, health-enhancing PA included autonomous exercise behavior (e.g., running, playing football) and unorganized and unstructured exercise behavior with low and moderate-intensity energy consumption (e.g., walking up and down stairs, performing housework) [50]. For autonomous exercise behavior, surveys have shown that high PA groups regularly participate in open skill sports (e.g., football, table tennis), which enhance the executive control network more than participating in closed skill sports (e.g., running, swimming) [51]. Based on the adaptive capacity model, Raichlen and Alexander [52] suggested that the relationship between cognition and PA depends on the adaption of the physiological system to stimuli, which emphasizes the role of cognitive stimuli during exercise. In simple terms, people who regularly engage in exercise with high cognitive demands (e.g., table tennis, football) are more likely to show improvements in attention than people who engage in exercise with low cognitive demands (e.g., walking, jogging) [51]. Thus, it will be necessary to consider the contribution of the type of sport or exercise to the effects on the executive control network in future studies.

Accumulating evidence suggests that college students are increasingly exhibiting sedentary behavior owing to their increasingly greater use of computers or smartphones [53,54]. In the present study, the time spent in sedentary behavior in the low PA group was much longer than that in the high PA group. Sedentary behavior not only is considered a risk factor for adverse health outcomes [55,56,57] but also is associated with cognitive function impairment [58,59]. Research also suggests that sedentary behavior is negatively correlated with executive control [60,61]. In addition to the difference between the high and low PA groups in the efficiency of the executive control network, we also found that the efficiency of the executive control network in the moderate PA group was higher than that in the low PA group. Given that the amount of time spent sitting as part of the sedentary behavior in the moderate PA group was longer than that in the low PA group, we speculate that the lower efficiency of executive control in the low PA group may be due to the longer time spent in sedentary behavior.

In the present study, the efficiency of the executive control network was positively correlated with PA level. The more PA an individual participated in, the greater executive control they possessed. This result is in agreement with that of the study by Stillman et al. [62], who found that individuals with higher PA levels experienced less interference in the Stroop task than individuals with lower PA. Although the amount of time spent sitting for the high PA group was shorter than that for the low PA group in the present study, on the whole, all three PA groups spent a substantial amount of time in sedentary behavior. The positive correlation between PA level and the efficiency of the executive control network further suggested that increased PA could improve the deleterious effect of sedentary behavior.

Our findings were inconsistent with those of Johnstone and Marí-Beffa [25], who found that the effect of physical exercise appeared to be solely on the alerting network, not on the orienting and executive control networks. This inconsistency may be due to the participants included in the two studies. In the study by Johnstone and Marí-Beffa [25], one group was composed of individuals with martial arts training experience, and the other group was composed of individuals with a given number of hours per week engaged in physical exercise (e.g., football, basketball) but not in martial arts training; thus, all participants could be considered physically active. This could explain, in part, why Johnstone and Marí-Beffa [25] did not find a significant difference in the executive control network between the martial arts group and the control group.

### 4.2. PA and the Alerting and Orienting Networks

Contrary to our expectations, the present study did not find a significant difference in the alerting and orienting networks for the three levels of PA assessed. Thus, our findings do not support those of previous research that suggested that PA had promoting effects on the alerting and orienting networks of attention [25,33,34]. This inconsistent result may be due to the difference in participants in the studies. For instance, the participants in a study by Noguera et al. [55] were older adults whose attention may have been impaired with age. This variable could have modulated the relationship between PA and attention such that the ability of PA to improve the alerting and orienting network functions may have been more pronounced in these participants [33,34]. We also found that although the participants in the present study were all young adults, the results varied depending on the type of exercise that the individuals regularly participated in. The participants in the study by Johnstone and Marí-Beffa [25] regularly participated in martial arts that are considered combat sports (e.g., karate, taekwondo, and judo) in which athletes face a rapidly changing scenario and must always watch out for threats from an opponent because the distance between an athlete and an opponent is very close. Therefore, athletes need to be on high alert in the face of threats to achieve good performance. Thus, the inconsistencies in the results among previous studies and ours suggest that the composition of the groups should be considered when examining the association between PA and attention networks.

In addition, methodological differences, such as the time between presentations of the cue and target as well as the validity of the cues to predict the target, may have also contributed to the inconsistencies in results between studies. Some previous studies have found that exercise facilitates the orienting network [63,64], but the stimulus onset asynchrony in those studies was variable and the cue did not always accurately predict the target. In the present study, the stimulus onset asynchrony was fixed and the spatial cue always predicted the target, making the ANT difficulty relatively low. Thus, our inability to detect a statistically significant difference in the effects of the three exercise groups on the alerting and orienting networks may be attributable to a ceiling effect.

The major strengths of the study included (1) using a multifunctional attentional task to investigate the attentional networks in young adults, and (2) assigning participants to one of three levels of PA (high, moderate, and low) based on their answers to the IPAQ, enabling us to identify which PA level was associated with better improvement on attentional networks in young adults. However, this study also had limitations. First, physical exercise in the present study was assessed by using the IPAQ. Despite the acceptable reliability and validity exhibited by the IPAQ and its extensive application in the physical exercise assessment of young Chinese adults, it remains a self-report instrument and is thus inherently subjective and relies on the participants of all the self-reported data. Objective measures of PA (e.g., accelerometer, body media) should be considered for future studies. Second, although our findings, as well as those of previous studies, indicated that physical exercise has selective benefits at the behavioral level among young adults, the underlying neural mechanisms were not explored and warrant further study. In future studies, event-related potentials and functional magnetic resonance imaging may be used to inform mechanistic understanding. Third, the present study used a cross-sectional design, which is an observational design and thus cannot determine whether a cause-and-effect relationship exists between PA and attentional networks. The use of a longitudinal study design in future explorations will facilitate the determination of a causal relationship between PA and attentional networks.

The present study found not only that PA promoted the executive control network in young adults but also that those who engaged in high or moderate PA showed superior executive control to those who engaged in only low PA levels. Our findings contribute to the knowledge regarding the effects of PA levels on attentional networks in young adults. In addition, our findings potentially provide useful information for policymakers and the public, suggesting that engaging in PA may enhance executive control. Encouraging college students, in particular, to reduce sedentary behavior and participate actively in physical exercise or sports club activities in their free time may help this population achieve psychological as well as physical benefits.

## 5. Conclusions

The present study contributed to our understanding of the association between PA and attention networks among Chinese young adults. Compared with participants in moderate and low PA groups, participants in a high PA group showed higher efficiency in the executive control network. By contrast, no significant differences were found among these three PA groups for the alerting and orienting networks. Our results provide supporting evidence for selective enhancement of PA on attentional networks, primarily in the executive control network, and suggest that PA may be a valuable approach for improving the executive control network in young adults.

## Figures and Tables

**Figure 1 ijerph-19-05374-f001:**
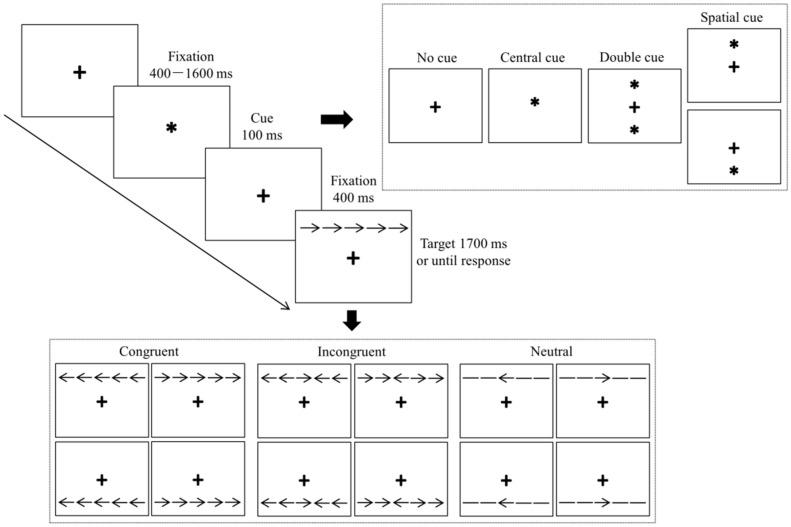
Sequence within a single trial of the attention network task.

**Figure 2 ijerph-19-05374-f002:**
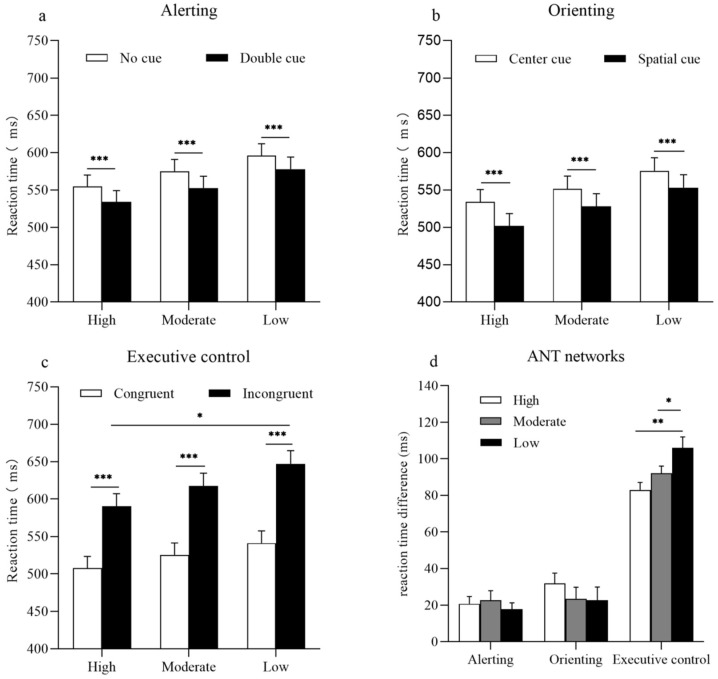
Reaction times for three levels of physical activity in each network. (**a**) For the alerting network, the bars represent mean reaction times as a function of an alerting effect and physical activity group. White bars represent reaction times for the no cue condition, and black bars represent reaction times for the double cue condition. (**b**) For the orienting network, the bars represent mean reaction times as a function of an orienting effect and physical activity group. White bars represent reaction times of the central cue condition, and black bars represent reaction times of the spatial cue condition. (**c**) For the executive control network, the bars represent mean reaction times as a function of a congruency effect and physical activity group. White bars represent reaction times for the congruent condition, and black bars represent reaction times for the incongruent condition. (**d**) Bars represent the reaction time differences between a specific cue or a flanking condition that reflect the efficiency of the Attention Network Test (ANT) networks for the high, moderate, and low PA groups. Error bars represent standard error of mean (SEM). * *p* < 0.05; ** *p* < 0.01; *** *p* < 0.001.

**Figure 3 ijerph-19-05374-f003:**
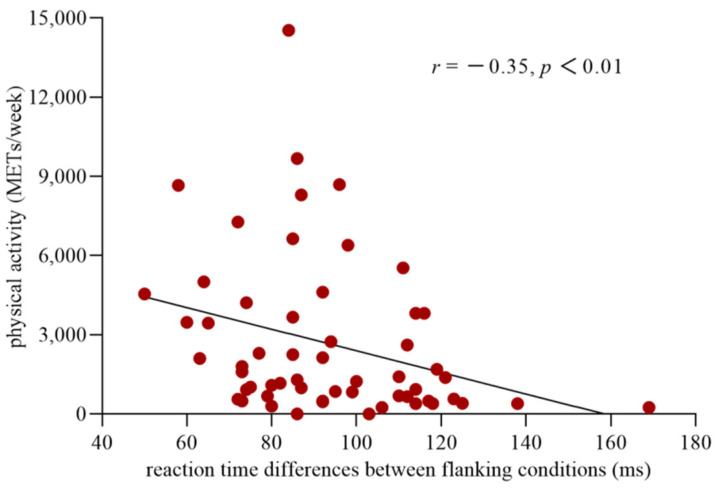
Correlation between the efficiency of the executive control network and the level of physical activity. The abscissa shows the reaction time differences between the incongruent and congruent conditions, which reflect the efficiency of executive control network, and the ordinate indicates the level of the physical activity.

**Table 1 ijerph-19-05374-t001:** Demographic characteristics of 57 included participants (mean ± standard error).

Characteristic	High PA	Moderate PA	Low PA
n	20	19	18
Female	8	8	7
Age (years)	20.10 ± 0.26	19.74 ± 0.23	19.83 ± 0.32
Height (cm)	173.55 ± 1.98	169.79 ± 1.81	172.39 ± 1.86
Body mass (kg)	63.95 ± 1.98	61.58 ± 3.10	64.17 ± 2.54
Body mass index (kg/m^2^)	21.46 ± 0.58	21.27 ± 0.94	21.49 ± 0.62
Education (years)	14.05 ± 0.22	13.74 ± 0.18	13.72 ± 0.19
IPAQ ^1^ (METs/week)	5843.85 ± 672.30	1465.50 ± 133.27	416.50 ± 47.40
Sitting time (min/day)	307.50 ± 131.62	412.63 ± 110.75	460.00 ± 151.23

^1^ IPAQ, International Physical Activity Questionnaire (short version in Chinese).

**Table 2 ijerph-19-05374-t002:** Mean reaction times on trials with correct responses in the Attention Network Test for groups with high, moderate, and low physical activity (mean ± standard error).

Group	Cue Type	Flanker Type		
		Congruent	Incongruent	Neural
High	No cue	538.14 ± 15.84	611.47 ± 17.42	516.73 ± 13.85
	Central cue	509.50 ± 17.24	601.29 ± 17.56	495.92 ± 15.43
	Double cue	507.29 ± 15.33	598.37 ± 17.37	498.96 ± 14.36
	Spatial cue	475.40 ± 16.29	552.42 ± 17.78	479.18 ± 16.70
Moderate	No cue	549.13 ± 16.25	639.63 ± 17.87	547.76 ± 14.21
	Central cue	522.25 ± 17.68	627.32 ± 18.01	516.14 ± 15.83
	Double cue	527.47 ± 15.73	621.66 ± 17.82	517.40 ± 14.73
	Spatial cue	502.34 ± 16.71	585.64 ± 18.24	500.86 ± 17.13
Low	No cue	561.84 ± 16.70	664.05 ± 18.36	566.76 ± 14.60
	Central cue	543.98 ± 18.17	649.15 ± 18.51	541.14 ± 16.27
	Double cue	540.29 ± 16.16	658.49 ± 18.31	541.72 ± 15.14
	Spatial cue	517.88 ± 17.17	618.64 ± 18.74	525.50 ± 17.60

**Table 3 ijerph-19-05374-t003:** Mean accuracy rates on the Attention Network Test for groups with high, moderate, and low physical activity (mean ± standard error).

Group	Cue	Flanker		
		Congruent	Incongruent	Neural
High	No cue	100.00 ± 0.25	94.05 ± 3.00	99.80 ± 0.29
	Central cue	99.40 ± 0.28	89.10 ± 3.17	99.80 ± 0.48
	Double cue	99.80 ± 0.23	94.35 ± 2.45	99.60 ± 0.39
	Spatial cue	100.00 ± 0.32	96.90 ± 2.22	99.60 ± 0.33
Moderate	No cue	99.79 ± 0.25	87.21 ± 3.08	98.95 ± 0.30
	Central cue	99.58 ± 0.29	84.74 ± 3.25	98.74 ± 0.49
	Double cue	99.79 ± 0.24	84.32 ± 2.52	99.58 ± 0.40
	Spatial cue	99.37 ± 0.33	91.32 ± 2.28	98.95 ± 0.34
Low	No cue	99.11 ± 0.26	90.61 ± 3.16	99.78 ± 0.30
	Central cue	99.78 ± 0.30	86.00 ± 3.34	98.44 ± 0.50
	Double cue	99.56 ± 0.25	90.06 ± 2.58	99.33 ± 0.42
	Spatial cue	99.33 ± 0.34	93.67 ± 2.34	100.00 ± 0.35

## Data Availability

The data presented in this study are available on request from the corresponding author.

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
