# Peer review of "Effects of Physical Activity Level on Attentional Networks in Young Adults"

_ijerph, 2022, doi:10.3390/ijerph19095374_

Round 1
Reviewer 1 Report
- Originality – the paper is related to a very interesting and not enough explored topic. However, the originality of the paper is not stated properly. Although it can be assumed what the main contribution of the manuscript is, its originality should be clearly emphasized in the abstract and especially in the introduction section. Besides of that, the theoretical and practical contribution of the paper are not mentioned enough. The contribution of the research should be clearly highlighted.
- Introduction and Literature review – The introduction section depicts in a fair way the context of the research and presents the key aspects of the topic needed for understanding the main aim of the research. However, as previously mentioned, it is very important to emphasize the originality of the paper in this part. The authors could do that by adding a single paragraph at the end of the introductory part. Given as now, the originality of the paper is not highlighted and the readers can only assume why the research is important i.e. what its main contribution is. The literature review covers quite wide range of the relevant literature. However, the quality of this section would benefit from adding more information regarding the effects of physical activity level on attentional networks in young adults in general. This part should be more elaborated and explained, as understanding of the current situation regarding the effects of physical activity level on attentional networks in young adults is also important for better understanding of the results and contribution of this research.
- Methodology – The methodology is explained in a clear manner.
- Results and Discussion – The results is explained in a clear manner. However, the discussion section should be reorganized. The discussion section should better emphasize how the obtained results can be used and why they are important. I strongly recommend the authors to add a paragraph explaining main theoretical and practical contribution of the paper, as this in not highlighted in the appropriate manner.
- Practical implications – The practical contribution of the paper is not clearly stated. It is not mentioned how the given results of the research can be helpful to the managers and other decision makers. The practical contribution of the paper should be reflected in providing decision makers with clear and accurate recommendations stemming from the research results, which can help them to make decisions in more effective and efficient way.
- Quality of Communication - The paper is written in a clear manner. The abbreviations are preceded by the full names of the terms which they represent, while the sentences are short enough and easy to understand.
Author Response
Response to Reviewer 1 Comments
Dear Reviewer #1,
We appreciate your time and consideration of this manuscript. The manuscript has been modified based on your useful recommendations. We have provided a point-by-point explanation of the changes made in response to the issues you raised, with changes pertaining to your comments labeled in red line.
We believe that this manuscript is now greatly improved due to the suggested changes. We hope that the changes will meet your approval. Again, thank you for the helpful comments, and we look forward to hearing from you soon.
The Authors
- Originality – the paper is related to a very interesting and not enough explored topic. However, the originality of the paper is not stated properly. Although it can be assumed what the main contribution of the manuscript is, its originality should be clearly emphasized in the abstract and especially in the introduction section. Besides of that, the theoretical and practical contribution of the paper are not mentioned enough. The contribution of the research should be clearly highlighted.
Response 1: Thank you for your comments. Again, the reviewer’s comments are much appreciated. We have added some content to the introduction and discussion to illustrate the originality of the paper (please see the revision on page 2, 3 & 10). In addition, we also have added some content to the latter part of the discussion to illustrate the theoretical and practical contribution of the paper (please see the revision on page 13).
- Introduction and Literature review – The introduction section depicts in a fair way the context of the research and presents the key aspects of the topic needed for understanding the main aim of the research. However, as previously mentioned, it is very important to emphasize the originality of the paper in this part. The authors could do that by adding a single paragraph at the end of the introductory part. Given as now, the originality of the paper is not highlighted and the readers can only assume why the research is important i.e. what its main contribution is. The literature review covers quite wide range of the relevant literature. However, the quality of this section would benefit from adding more information regarding the effects of physical activity level on attentional networks in young adults in general. This part should be more elaborated and explained, as understanding of the current situation regarding the effects of physical activity level on attentional networks in young adults is also important for better understanding of the results and contribution of this research.
Response 2: We appreciate the reviewer’s constructive suggestions. Following your suggestion, we have added some content to the last paragraph of the introduction to illustrate the originality of the paper (please see the revision on page 2, 3 & 10).
In addition, there has been little previous research on the effect of PA levels on attentional network in young adults. To be best of our knowledge, this is the first attempt to examine the influence of PA levels on attentional networks via ANT in young adults. Thus, we added some researches about the effect of PA on attentional networks in other population in order to facilitate readers to better understand our study (please see the revision page 2).
- Methodology – The methodology is explained in a clear manner.
Response 3: Thank you very much for your review.
- Results and Discussion – The results is explained in a clear manner. However, the discussion section should be reorganized. The discussion section should better emphasize how the obtained results can be used and why they are important. I strongly recommend the authors to add a paragraph explaining main theoretical and practical contribution of the paper, as this in not highlighted in the appropriate manner.
Response 4: Thank you for your comments. Following your suggestion, we have added some content to the latter part of the discussion to illustrate the theoretical and practical contribution of the paper (please see the revision on page 13).
- Practical implications – The practical contribution of the paper is not clearly stated. It is not mentioned how the given results of the research can be helpful to the managers and other decision makers. The practical contribution of the paper should be reflected in providing decision makers with clear and accurate recommendations stemming from the research results, which can help them to make decisions in more effective and efficient way.
Response 5: Thank you for your comments. Following your suggestion, we have added some content to the latter part of the discussion to illustrate the contribution of the paper (please see the revision on page 13).
- Quality of Communication - The paper is written in a clear manner. The abbreviations are preceded by the full names of the terms which they represent, while the sentences are short enough and easy to understand.
Response 6: Thank you very much for your review.

Reviewer 2 Report
Dear AUTHORS:
Consider these minor details in order to improve the final version
King regards

Author Response
Response to Reviewer 2 Comments
Dear Reviewer #2,
We appreciate your time and consideration of this manuscript. The manuscript has been modified based on your useful recommendations. We have provided a point-by-point explanation of the changes made in response to the issues you raised, with changes pertaining to your comments labeled in blue.
We believe that this manuscript is now greatly improved due to the suggested changes. We hope that the changes will meet your approval. Again, thank you for the helpful comments, and we look forward to hearing from you soon.
The Authors
- On page 1, line 22; ‘among these three groups’
Response 1: Thank you for your suggestion. We have changed “between” to “among” (please see the revision on page 1).
- On page 1, line 31; ‘it’s’
Response 2: Thank you for your suggestion. We have changed “It is” to “It’s” (please see the revision on page 1).
- On page 1, line 39; such as college students. Here, I would include for the best of the authors knowledge.
Response 3: Thank you for your suggestion. We added some content about the importance of attention to college students (please see the revision on page 1). We wonder if we have accurately understood the meaning of the reviewer.
- On page 1, line 41; The aim of the present study was to investigate whether the positive effects of PA on attention observed in other populations extend to young adults. Is the sentence correct here?
Response 4: Thank you for your suggestion. We have moved this sentence to the last paragraph of introdution (please see the revision on page 2 & 3).
- On page 3, line 120; the reference with Fortaleza actualization should be included.
Response 5: Thank you for your comment. The reference with Fortaleza actualization has been added (please see the revision on page 3 &15).
- On page 4, line 160; Did you check the normality of the data?
Response 6: Thank you for raising this issue. We had checked the normality of the data before submitting the manuscript. Most of data conformed to the normal distribution, only a small fraction of the data was close to normal distribution. The reason for these results might be due to the extreme values rejection method used in our study, i.e., reaction time below 200 ms and above 1700 ms were excluded from further analysis. We found that there were several extreme values affected the normality of the data through the histogram and Q-Q plots. In addition, our study mainly focused on the pairwise comparison between groups, which didn’t constrain strictly on the normality of the data. Thus, we believed that our results were reliable.
- On page 4, line 163; Did you check the Levene test?
Response 7: The reviewer’s comments are much appreciated. We didn’t check the Levene test before submitting the manuscript. The results of Levene test revealed that the variance between groups was homogeneous on gender, age, height, body mass, body mass index and the amount of time spent sitting, while the variance between groups was not homogeneous on PA level and years of education. The intergroup variation was analyzed by one-way analysis of variance (ANOVA) followed by Least Significant Difference test (LSD) when variances were homogeneous or Dunnetts T3 test when variances were not homogeneous. Thus, we have revised the results of PA level (please see the revision on page 5).
- On page 4, line 165; Include Fergusson.
Response 8: Thank you for your suggestion. We are sorry that we don’t understand the meaning of Fergusson.
- On page 4, line 167; Post hoc..? LSD …… Include in statistical part?
Response 9: Thank you for your comment. An Dunnetts T3 post-hoc comparison and An LSD post-hoc comparison were included in the results part (please see the revision on page 5).
- On page 4, line 166; ‘Body mass’
Response 10: Thank you for your suggestion. We have changed “weight” to “body mass” (please see the revision on page 5).
- On page 4,5, line 173,182; ‘between-participants’
Response 11: Thank you for your suggestion. We have changed “subjects” to “participants” throughout the text (please see the revision on page 5).
- On page 5, line 196; Did you check the magnitude of correlation?
Response 12: Thank you for your comments. Following your suggestion, we have added the magnitude of correlation to the correlation between executive control and PA level (please see the revision on page 10 & 15).
- On page 5, line 203; ‘Body mass’
Response 13: Thank you for your suggestion. We have changed “weight” to “body mass” (please seethe revision on page 5).
- On page 5, line 213; table 1, ‘Body mass’
Response 14: Thank you for your suggestion. We have changed “Weight” to “Body mass” (please see the revision on page 6).
- On page 8, line 285-297, 324; Figure 2, 3, Font (letter Little)
Response 15: Thank you for your comment. The front size and word size were consistent with the template in Figure 2, 3.
- On page 9, line 331; Young adults, define better this word trought the text
Response 16: Thank you for your comment. We have deleted the word “healthy”, and used “young adults” throughout the text (please see the revision on page 1 & 10).
- On page 10, line 373; In simple terms, people who regularly engage in exercise with high cognitive demands (e.g., table tennis, football) are more likely than people who engage in exercise with low cognitive demands (e.g., walking, jogging) to show improvements in attention. Agree, but include one reference why..?
Response 17: Thank you for your comment. Following your suggestion, we have now added a reference to support this statement in the text (please see the revision on page 11).
- On page 11, line 428; ‘among previous studies’
Response 18: Thank you for your suggestion. We have changed “between” to “among” (please see the revision on page 12).
- On page 11, line 440; Agree with the asumpption of limitation, but in your study there some strenghts:
Response 19: Thank you for your suggestion. We have add some strenghts of the study before the limitations (please see the revision on page 12).
- On page 11, line 443; ‘participants’
Response 20: Thank you for your suggestion. We have changed “subject” to “participants” (please see the revision on page 13).
- On page 10, line 352; higher health-enhancing PA and lower sedentary behavior might explain the superiority of the efficiency of the executive control network. Agree with this sentence but I think it needs a reference supporting it.
Response 21: Thank you for your comments. Again, the reviewer’s comments are much appreciated. We have added two references to support this statement (please see the revision on page 11 & 15).
